# Comprehensive 3DCRT Hypofractionated Radiotherapy Schedule for Localized Prostate Adenocarcinoma in the Era of IMRT: Dosimetric and Endoscopic Analysis

**DOI:** 10.3390/cancers16061192

**Published:** 2024-03-18

**Authors:** Andromachi Kougioumtzopoulou, Nick Syrigos, Anna Zygogianni, Ioannis Georgakopoulos, Kalliopi Platoni, George Patatoukas, Kimon Tzannis, Aristotelis Bamias, Nikolaos Kelekis, Vasileios Kouloulias

**Affiliations:** 1Radiotherapy Unit, 2nd Department of Radiology, ATTIKON University Hospital, Medical School, National & Kapodistrian University of Athens, 12462 Athens, Greece; kplatoni@med.uoa.gr (K.P.); gpatatouk@med.uoa.gr (G.P.); kelnik@med.uoa.gr (N.K.); vkouloul@med.uoa.gr (V.K.); 2Oncology Unit, 3rd Department of Internal Medicine, Medical School, National & Kapodistrian University of Athens, 12462 Athens, Greece; nksyrigos@med.uoa.gr; 31st Department of Radiology, Radiotherapy Unit, Medical School, National and Kapodistrian University of Athens, 12462 Athens, Greece; azygogianni@med.uoa.gr (A.Z.); ioangeo@med.uoa.gr (I.G.); 4Second Propaedeutic Department of Internal Medicine, ‘Attikon’ University Hospital, “National & Kapodistrian University of Athens, 12462 Chaidari, Greece; kimon.tzannis@gmail.com (K.T.); abamias@med.uoa.gr (A.B.)

**Keywords:** moderate hypofractionated radiotherapy, prostate cancer, intensity-modulated radiation therapy (IMRT), three-dimensional conformal radiation therapy (3DCRT), dosimetry, rectoscopy, rectal wall, Vienna Rectoscopy Score (VRS), dose volume histogram (DVH)

## Abstract

**Simple Summary:**

Moderate hypofractionated radiotherapy (MHRT) has emerged as the preferred treatment modality for localized prostate cancer. The aim of this prospective study was to evaluate late rectal toxicity while performing a comprehensive dosimetric analysis in conjunction with rectoscopy results in the setting of MHRT. We confirmed in an homogeneous population (20 patients) that when 3DCRT is employed, delineation of rectal wall and its subsegments can provide a more in depth accurate dosimetric analysis. Furthermore, we identified that dose endpoints V52.17Gy and V56.52Gy, when hypofractionated schedule (2.75 Gy per fraction) is used, may have a significant impact on rectal mucosal injury. Implementing these variables in clinical practice may result in a decrease rate of late rectal toxicity; more data is needed to assist in further validation of this conclusion.

**Abstract:**

**Background:** Moderate hypofractionated radiotherapy (MHRT) has emerged as the preferred treatment modality for localized prostate cancer based on randomized controlled studies regarding efficacy and toxicity using contemporary radiotherapy techniques. In the setting of MHRT, available data on dosimetric parameters and late rectal toxicity are limited. **Aim:** To present the effects of MHRT on late rectal toxicity while conducting an extensive dosimetric analysis in conjunction with rectoscopy results. **Methods:** This is a prospective study including patients with intermediate-risk prostate adenocarcinoma. All patients were treated with MHRT 44 Gy in 16 fractions to the seminal vesicles and to the prostate, followed by a sequential boost to the prostate alone of 16.5 Gy in 6 fractions delivered with three-dimensional conformal radiation therapy (3DCRT). Acute and late toxicity were assessed. Endoscopy was performed at baseline, every 3 months post-therapy for the first year, and every 6 months for the year after. The Vienna Rectoscopy Score (VRS) was used to assess rectal mucosal injury related to radiotherapy. Dosimetric analysis for the rectum, rectal wall, and its subsegments (upper, mid, and low 1/3) was performed. **Results:** Between September 2015 and December 2019, 20 patients enrolled. Grade 1 late gastrointestinal toxicity occurred in 10% of the patients, whereas 5% had a grade ≥2. Twelve months post radiotherapy: 4 (20%) patients had VRS 1; 2 (10%) patients had VRS 2; 1(5%) patient had VRS 3. 24 months post radiotherapy, VRS 1 was observed in 4 patients (20%) and VRS 2 in 3 (15%) patients. The dosimetric analysis demonstrated noticeable variations between the rectum, rectal wall, and rectal wall subsegments. The dosimetric analysis of the rectum, rectal wall, and its mid and low segments with respect to rectoscopy findings showed that the higher dose endpoints V52.17_Gy_ and V56.52_Gy_ are associated with rectal mucosal injury. **Conclusions:** A thorough delineation of the rectal wall and its subsegments, together with the dosimetric analysis of these structures, may reduce late rectal toxicity. Dosimetric parameters such as V52.17_Gy_ and V56.52_Gy_ were identified to have a significant impact on rectal mucosal injury; additional dose endpoint validation and its relation to late GI toxicity is needed.

## 1. Introduction

Moderate hypofractionated radiotherapy (MHRT) in localized prostate cancer has emerged as the standard of care regarding radiotherapy treatment [1]. Several phase III randomized controlled trials (RCTs) provide evidence in support of the use of MHRT compared to conventionally fractionated radiotherapy when contemporary radiotherapy techniques, such as intensity-modulated radiation therapy (IMRT) and three-dimensional conformal radiation therapy (3DCRT) are employed [2,3,4,5,6,7,8,9,10].

Prostate adenocarcinoma has been considered an ideal malignancy for MHRT owing to its notably low α/β ratio, as viewed from a radiobiological point of view. There is a growing body of empirical data that supports that prostate cancer possesses an α/β ratio of 1.5 Gy (with a range of 0.9–2.2 Gy), which is exceptionally low [11]. Additionally, the adjacent tissues and organs at risk (OARs), such as the rectum and bladder, exhibit a higher α/β ratio, typically within the range of 3–7 Gy [12,13]. The impact of prostate cancer has a lower a/b ratio in comparison to adjacent normal tissue and has the potential to enhance the therapeutic ratio through the utilization of hypofractionated schedules. Ideally, this could result in enhanced tumor control and a low incidence rate of toxicity. In spite of advancements in technology and a favorable radiobiological perspective, the rectum continues to be a dose-limiting organ in prostate external beam radiotherapy, including MHRT [14,15,16].

A number of studies have demonstrated a relationship between parameters derived from the dose-volume histogram (DVH) and the incidence of rectal toxicity. Nevertheless, the available literature on the setting of MHRT and its association with late rectal toxicity is scarce, as evidenced by the limited data exploring the correlation between specific dose/volume metrics and late rectal adverse events [16,17,18,19,20,21,22]. In addition, various schedules of MHRT have been examined, and different parameters related to dose volume have been evaluated.

Rectosigmoidoscopy is a useful diagnostic procedure that offers an accurate and thorough assessment of rectal mucosal injury related to irradiation by recognizing pre-existing pathological problems and detecting tissue changes that may not yet be apparent through subjective symptoms alone [23]. The Vienna Rectoscopy Score (VRS) is a validated tool utilized for the assessment of rectal mucosal injury based on the endoscopic terminology of the World Organization for Digestive Endoscopy resulting from pelvic irradiation [23,24,25,26].

It has been shown that rectosigmoidoscopy performed one year after pelvic RT, using VRS to evaluate the rectal mucosa, can predict late-onset rectal toxicity and thus may serve as a surrogate endpoint [27]. Currently, scarce data exist on rectal mucosal injury related to radiotherapy in the setting of the hypofractionation [28]. In addition, rectal injury after pelvic radiotherapy could be related to other clinical patient-specific factors such as age, diabetes, and anticoagulant drugs [29,30,31,32]. Rectosigmoidoscopy findings have been reported elsewhere but have not been correlated with late toxicity or dosimetric factors [33].

Considering the aforementioned parameters, the aim of this study was to assess the incidence of late rectal toxicity while additionally performing a comprehensive dosimetric analysis in correlation with rectoscopy findings in patients treated with MHRT for localized prostate cancer. This work provides updated results regarding late toxicity and aims to investigate the potential predictive role of rectal dosimetric parameters on rectal mucosal changes after RT.

## 2. Materials & Methods

### 2.1. Study Design and Motivation

A prospective cohort study was conducted in patients with intermediate-risk prostate cancer undergoing radical radiotherapy. The study protocol received approval from the scientific and ethical committee of the institutional review board (N:21107). The rationale of this study was to assess the feasibility of employing MHRT using 3-DCRT in Greece through the national healthcare system, given the limited availability until recently of contemporary radiotherapy techniques such as intensity-modulated radiation therapy (IMRT) for a substantial percentage of the patient population, a situation not uncommon in many countries globally as it has been reported from the International Atomic Energy Agency (IAEA) [34,35].

### 2.2. Patients

Patients having a histologically confirmed diagnosis of prostate adenocarcinoma with intermediate risk characteristics above the age of eighteen were eligible. Patients who met any of the following criteria were categorized into the intermediate-risk group: clinical stage T1-T2cN0M0, Gleason score ≤ 7, (grade group 1–3) and PSA level 10–20 ng/mL (as per to American Joint Committee on Cancer staging system, 7th edition) [36]. Patients with high- or very high-risk features and with clinical evidence of lymph node involvement and/or metastases were excluded from the study. Other exclusion criteria for this study included any prior treatment for prostate cancer, any malignancy diagnosed within the previous five years (excluding nonmelanoma skin cancer), any known inflammatory bowel disease, any prior pelvic irradiation, or any contraindication to radiotherapy.

Prior to the initiation of treatment, a comprehensive physical examination was conducted, encompassing an evaluation of genitourinary (GU) and gastrointestinal (GI) complaints. Prior to initiating treatment, it was necessary to conduct a baseline colonoscopy on each participant included in the study. The administration of neoadjuvant and/or adjuvant androgen deprivation therapy (ADT) was implemented for all patients for a short duration of 4–6 months, both preceding and throughout the course of radiotherapy. Before starting treatment, all study participants provided written informed consent.

### 2.3. Radiotherapy

Each patient in the study underwent a computer tomography (CT) planning scan, with slices taken at 3 mm intervals, in supine position using a feet rest devise, with a full bladder and an empty bowel. All participants were scheduled to receive MHRT within 6 weeks of registration. A total dose of 60.5 Gy given in 22 fractions, in five fractions per week, over 4 weeks was prescribed in all study individuals.

The first clinical target volume (CTV1) was defined as the prostate with seminal vesicles as it was at the time of CT simulation. The first planning target volume PTV1 was created by a non-uniform expansion of the CTV1 of 6mm posteriorly and 10mm in all the rest directions to a total dose of 44 Gy in 16 fractions prescribed to PTV1. The second clinical target volume (CTV2) was defined as the prostate alone, and the subsequent planning target volume (PTV2) was created by a non-uniform expansion of the CTV2 of 6mm posteriorly and 10mm in all the rest directions; a total dose of 16.5 Gy in 6 fractions was prescribed to PTV2. The rationale to include the seminal vesicles in CTV1 was based on the Partin staging nomogram, where the probability of seminal vesicle invasion is 15% for patients with one risk factor (PSA over 10, Gleason over 6, T-stage over T2a) and the prescription to the seminal vesicles and to the prostate followed the CHHip protocol for intermediate-risk cancer [7,37]. Based on the linear quadratic (LQ) model with α/β ratio of 1.5 Gy for the prostate cancer and the seminal vesicles, the estimated equivalent total dose (EQD2) for the prostate was 73.4 Gy and 53.4 Gy for the seminal vesicles [38].

The femoral heads, bladder, and rectum were delineated following the relative EORTC recommendations [39]. Additionally, the rectal wall and its sub-volumes, which encompassed the upper, middle, and lower 1/3 segment of the rectal wall, were delineated for the purpose of this study. Delineation of the rectal wall was performed manually, followed by automatic addition of an inner and outer margin of 1 mm to delineate the rectal wall volume, resulting in a rectal wall thickness between 2 and 3mm. Dose constraints to organs at risk in this study protocol were in accordance with Quantitative Analyzes of Normal Tissue Effects in the Clinic (QUANTEC) guidelines [40]. EQD2 was used, assuming an α/β ratio of 3 Gy for late rectal toxicity as reported in study protocol Radiation Therapy Oncology Group (RTOG) 0126 [41]. Late radiation-induced toxicity was evaluated using both the Late Effects Normal Tissue Task Force/Subjective, Objective, Management, Analytic (LENT/SOMA) of the European Organization for Research and Treatment of Cancer/Radiation Therapy Oncology Group score (EORTC/RTOG) and the Subjective -RectoSigmoid (S-RS) scale [42,43,44]. In addition, dose volume histograms (DVH) for all patients were recorded for the rectum, rectal wall, and its subvolumes for the following dose endpoints: V26.08_Gy_, V34.78_Gy_, V43.48_Gy_, V52.17_Gy_, V56.52_Gy_ and V60.87_Gy_.

3DCRT irradiation with the utilization of image-guidance was performed with either daily megavoltage portal images (MV portals) or with kilovoltage cone-beam CT (CBCT) before each treatment. The treatment was delivered via linear accelerators with a beam energy of 10 or 15 MV. All patients in the study were advised to follow a low fat and low fiber diet during treatment.

### 2.4. Toxicity and Follow-Up

Toxicity assessment by a physician was performed weekly during treatment, at the completion of radiotherapy treatment, every 3 months for the first year, and every 6 months for the next 5 years. Acute radiation-related toxicity was considered as any adverse event that occurred during treatment and for a period of 6 months after the end of treatment. Common Terminology Criteria for Adverse Events (CTCAE v4.0) was used to evaluate acute toxicity. Late radiation-induced toxicity was evaluated using the Late Effects Normal Tissue Task Force/Subjective, Objective, Management, Analytic (LENT/SOMA) and the Subjective-RectoSigmoid (S-RS) scale [37,38,39]. Late radiation toxicities were defined as any adverse event occurring 6 months from the completion of treatment and beyond. Based on our study protocol, each patient had a colonoscopy before radiotherapy and every 3 months after the completion of treatment for the first year. During the second year, a rectoscopy was performed every 6 months. The endoscopic procedure was performed using a video colonoscope. The endoscopic findings were reported according to the World Organization for Digestive Endoscopy and were evaluated based on the Vienna Rectoscopy Score (VRS). Biochemical recurrence following radiotherapy was defined as per the Phoenix criteria [45].

### 2.5. Endpoints

The primary endpoint of the study protocol was late rectal toxicity. Furthermore, this study aimed to correlate dosimetric parameters and rectoscopy findings with the incidence of adverse GI events in late GI. Secondary endpoints included acute GI adverse events, acute and late GU toxicity, evaluation of late rectal toxicity scales, and dosimetric analysis of the rectum, rectal wall, and its subvolumes. Finally, we investigated a potential correlation between the dose-volume DVH parameters of the rectum, rectal wall, and rectal wall subvolumes with the rectoscopy findings. The assessment of treatment effectiveness involved the evaluation of biochemical failure, which was determined according to the Phoenix criteria.

### 2.6. Statistical Analysis

Continuous variables were summarized with the use of descriptive statistical measures (median and IQR [interquartile range]), and categorical variables were displayed as frequency tables (N, %). The normality of continuous variables was evaluated using the Shapiro–Wilk test and graphical methods. Fisher exact test and Pearson chi-squared test were used to compare categorical data, and Wilcoxon rank sum tests or t-tests were used to compare continuous variables among groups of patients. Box plots were used to graphically demonstrate the symmetry, skew, variance, and outliers of our data and to visually compare them between groups. All statistical tests were two-sided and were performed at a 0.05 significance level. All statistical analyses were performed using the statistical package STATA (StataCorp. 2023. Stata Statistical Software: Release 18. College Station, TX, USA: StataCorp LLC).

## 3. Results

### 3.1. Patients’ Characteristics

Twenty patients with prostate adenocarcinoma were treated between September 2015 and December 2019 according to the study protocol and entered the analysis. The median follow-up time was 4.7 years, with an interquartile range of 1.8 years. The demographics and clinical characteristics are summarized in Table 1. At the time of diagnosis, the median age was 72.7 years with an interquartile range of 6.3 years, and the median value of PSA was 7.4 ng/mL with an interquartile range of 12.6 ng/mL. Benign findings in colonoscopy were found in 10 (50%) study patients before starting radiotherapy. All patients received the study hypofractionated schedule without interruptions.

### 3.2. Toxicity

#### 3.2.1. Acute Toxicity

Acute adverse events regarding bowel and bladder are summarised in Table 2 and Table 3, respectively.

Throughout the acute period, 40% of the patients had grade 1 GI adverse events, whereas 15% had grade 2. During the first two weeks of the radiotherapy treatment, no radiation-induced toxicity was reported. In the third week of radiotherapy, grade 1 GI-adverse events occurred in 6 (30%) patients, and only 1 patient (5%) experienced grade 2 toxicity. At the fourth week of treatment, grade 1 GI adverse events were seen in 3 patients (15%), while 2 patients (10%) had grade 2 related adverse events. At the end of treatment, 2 study patients (10%) experienced grade 1 GI-adverse events. At 3 and 6 months after treatment, no GI-adverse events occurred. No grade 3–4 GI adverse events were observed.

Grade 1 acute GU-adverse events occurred as follows: 1 patient (5%) at the first week of radiotherapy; 2 (10%) patients at the second week of radiotherapy; 8 (40%) patients at the third week of radiotherapy; 11 (55%) patients at the fourth week of radiotherapy; 10 (50%) patients at the end of treatment; and 1 patient (5%) at 6 months post-radiotherapy. Acute GU toxicity of grade 2 or higher did not occur in any patient.

#### 3.2.2. Late Toxicity

Grade 1 delayed GI adverse events occurred in 10% of patients, whereas 5% had grade 2 or higher. Late GI adverse events are shown in Table 4. At 9 months post radiotherapy, none of the study patients experienced any toxicity. At 12 months post radiotherapy, grade 1 and grade 3 GI toxicity occurred in 1 (5%) patient and in 1 (5%) patient, respectively. One patient experienced grade 3 late GI 12 months after completing radiotherapy, presenting rectal bleeding that required hospitalization and modest intervention. At 18- and 24 months post radiotherapy, grade 2 GI adverse events occurred in 1 patient (5%) and one patient (5%), respectively. No grade 4 GI-related toxicity was observed.

Late GI toxicity was assessed using both the LENT/SOMA and S-RS scales and there was no significant difference between the two scales; instead, there was an exact match (*p* > 0.99).

Late GU adverse events are shown in Table 5. Grade 1 late GU-adverse events occurred as follows: 1 patient (5%) at 9- and 12-months after radiotherapy; 2 (10%) patients at 18 months after radiotherapy; and 6 (30%) patients at 24 months after radiotherapy. No grade 2 or higher adverse events were observed.

All patients underwent rectoscopy according to the study protocol, and the relative findings according to VRS are summarized in Table 6. At 3 months post-treatment, there was no mucosal rectal damage in any of the patients. At 6 months after radiotherapy, 2 (10%) of the patients had mucosal rectal damage of VRS 2. At 9 months after radiotherapy, 2 (10%) of the patients had mucosal rectal damage of VRS 2 (the same 2 patients as reported at 6 months post radiotherapy). At 12 months after radiotherapy, mucosal rectal damage according to VRS observed was as follows: 4 (20%) patients had VRS 1; 2 (10%) patients had VRS 2; 1 (5%) patient had VRS 3. At 18 and 24 months after radiotherapy, mucosal findings of VRS 1 and VRS 2 were observed in 4 (20%) and 3 (15%) patients, respectively. Typical rectoscopy images from two study patients before treatment and 12 months after RT with a VRS 1 rectal mucosal lesion are shown in Figure 1.

Statistical analysis was performed to compare the late GI-related adverse events according to the LENT-SOMA scale with the rectoscopy findings evaluated according to the VRS, as shown in Table 7. At 3 months after radiotherapy, none of the patients had any related toxicity and no mucosal damage to the rectum. Two cases (the same patients, numbered 3 and 9) showed inconsistency at 6 and 9 months after radiation; they had signs of mucosal damage in rectoscopy, although there were no reported GI-related adverse events (*p* = 0.16). Indication of discrepancy at 12 months after radiotherapy was detected between the VRS findings and the LENT-SOMA scale for the 5 suggestive patients who had mucosal damage but no related toxicity (*p* = 0.063). A statistically significant difference was observed between the findings of the rectoscopy and the LENT-SOMA scale at 18 and 24 months after radiation therapy for the same six patients. These patients had mucosal damage but no associated subjected toxicity (*p* = 0.031).

### 3.3. Dosimetric Analysis

Assuming an α/β ratio of 3 Gy for late toxicity of the rectum, EQD2 was calculated and evaluated using the QUANTEC dose-volume constrains shown in Table 8. Even though, there are no QUANTEC constraints regarding doses below 50 Gy for the rectum, we included and analyzed two such dosimetric constraints in our study; more in particular we considered the V30_Gy_ and V40_Gy_ dose endpoints.

Dosimetric analysis was performed for the following volumes: rectum, rectal wall, the upper 1/3 of rectal wall, the mid 1/3 of rectal wall and the lower 1/3 of rectal wall. Rectal wall and rectal wall subvolumes were compared with the complete rectum volume in terms of dosimetry as seen in Table 8.

For dose endpoints of the rectum compared to the rectal wall, the difference was more notable in higher doses: V52.17_Gy_: 17.2% vs. 22.6% (*p* < 0.001); V56.52_Gy_: 11% vs. 15.9% (*p* < 0.001); V60.87_Gy_: 0.2% vs. 1.3% (*p* < 0.001). Likewise, for dose endpoints of the rectum compared to the upper 1/3 of the rectal wall, the difference was most evident in higher doses: V52.17_Gy_: 17.2% vs. 5.5% (*p* < 0.001); V56.52_Gy_: 11% vs. 1.1% (*p* < 0.001); V60.87_Gy_: 0.2% vs. 0% (*p* < 0.001). In contrast, for dose endpoints of rectum compared to mid 1/3 of rectal wall the difference was more pronounced at almost all doses: V26.08_Gy_: 89.2% vs. 100% (*p* < 0.001); V43.48_Gy_: 32.3% vs. 46.3% (*p* < 0.001); V52.17_Gy_: 17.2% vs. 34.7% (*p* < 0.001); V56.52_Gy_: 11% vs. 29.4% (*p* < 0.001). Finally, for dose endpoints of the rectum compared to the lower 1/3 of the rectal wall, the difference was most clear only in two doses: V52.17_Gy_: 17.2% vs. 29.3% (*p* < 0.001); V56.52_Gy_: 11% vs. 22.7% (*p* < 0.001). All investigated structures had a statistically significant difference in higher dose endpoints V52.17_Gy_ and V56.52_Gy_, as seen in Figure 2 and Figure 3.

### 3.4. Dosimetry and Rectoscopy

Analysis of the DVH of the rectum, rectal wall, and its subvolumes with respect to rectoscopy findings was performed. The analysis of dose endpoints of the rectum, rectal wall, upper 1/3, mid 1/3, and lower 1/3 of the rectal wall compared to the findings of rectoscopy are presented in Table 9, Table 10, Table 11, Table 12, and Table 13, respectively. Group A refers to patients within normal limits of rectal mucosa (VRS 0). Group B refers to patients with any sign of rectal mucosal damage observed (VRS 1, 2, or 3). At 3 months after radiotherapy, all patients had VRS within normal limits. Both Groups A and B remained stable at 6- and 9-month post-radiotherapy; thus, the data for the two time periods are identical. The same fact applies to the periods of 12, 18, and 24 months following the completion of radiotherapy.

Regarding the rectum, at 6 and 9 months after radiotherapy, a significant difference was noted for dose endpoints of V34.78_Gy_ and V43.48_Gy_; 47% vs. 68.8% (*p* = 0.042) and 30.9% vs. 48.6% (*p* = 0.042) between the two groups. At 12, 18 and 24 months after radiotherapy, significant difference was found for rectal dose endpoints of V34.78_Gy_ (40% vs. 61.8%, *p* = 0.009), V43.48_Gy_ (27.6% vs. 45.6%, *p* = 0.006), V52.17_Gy_ (11.8% vs. 27.8%, *p* = 0.006) and V56.52_Gy_ (7.4% vs. 20.7%, *p* = 0.006) for the two study groups.

With regards to the rectal wall, a significant difference was observed at 12, 18 and 24 months after radiotherapy, for dose endpoints of V34.78_Gy_ (42.8% vs. 55.5%, *p* = 0.005), V43.48_Gy_ (33.2% vs. 42.3%, *p* = 0.005), V52.17_Gy_ (19.9% vs. 30.9%, *p* = 0.005), V56.52_Gy_ (15.7% vs. 24.7%, *p* = 0.046) and V60.87_Gy_ (0.003% vs. 7.8%, *p* = 0.033) between group A and B.

Analysis for rectal wall subsegments revealed the following: for the upper 1/3, no significant difference was found for any dose endpoint during the follow-up period between the study groups; for the mid 1/3, a significant difference was noted at 12, 18 and 24 months after radiotherapy, for dose endpoints of V60.87_Gy_ (0.0% vs. 3.1%, *p* = 0.034) between group A and group B; for the lower 1/3 significant difference was observed at 12, 18 and 24 months after radiotherapy, for dose endpoints of V34.78_Gy_ (44.7% vs. 62.5%, *p* = 0.005), V43.48_Gy_ (33.6% vs. 52.3%, *p* = 0.002), V52.17_Gy_ (26.8% vs. 44.5%, *p* = 0.003), V56.52_Gy_ (18.6% vs. 36.4%, *p* = 0.006) and V60.87_Gy_ (0.0% vs. 9.1%, *p* = 0.015) for the two study groups.

### 3.5. Biochemical Recurrence

At the time of analysis and after a 5-year follow-up period, only 1 patient experienced biochemical recurrence from the 20 included patients. Biochemical recurrence occurred in the fourth year after the completion of radiotherapy.

## 4. Discussion

To the best of our knowledge, this is the first prospective study assessing the incidence of late rectal toxicity with the addition of performing a comprehensive dosimetric analysis in correlation with rectoscopy findings in a cohort of patients undergoing radical MHRT for intermediate-risk adenocarcinoma. Until 2017, the only radiotherapy technique available in our department was 3DCRT, and this technique was employed for all our patients.

Regarding acute toxicity, we report that GI adverse events were mild (15% had grade 2), and no grade 3–4 adverse events occurred in any of the patients. Acute GU toxicity of grade 2 or higher did not occur in any patient. The findings of our study are consistent with prior phase II prospective studies that focused on the use of MHRT in localized prostate cancer. A recent review reports that grade 2 acute GI ranged between 9 and 36% and grade ≥ 3 adverse events were rare, while acute GU grade 2 ranged between 7 and 39.1% and grade ≥ 3 adverse events were ≤5% [46].

Concerning late toxicity, we report that 5% of patients experienced late GI adverse events of grade 2 or higher, and none had late GU adverse events of grade 2 or higher. These findings are consistent with the outcomes reported in phase III RCTs dealing with MHRT for localized prostate cancer in which the 3DCRT technique was used [2,3,6,9,10].

The reason behind the investigation of the rectal wall and its subsegments was based on the fact that the use of the DVH on “hollow” organs, such as the bladder and rectum, could be considered controversial because it implies that these are solid organs. From a radiobiological perspective, it is the rectal wall rather than its contents that is considered the critical structure when assessing potential relative risks. Ultrasound data indicate a median rectal wall thickness of 2.6 mm, supporting the 3 mm wall thickness chosen in our study [47]. Tucker et al. noticed negligible changes in the dose wall DVH throughout rectal walls ranging from 2 mm to 5 mm [48]. Guckenberger et al. recommend semiautomatic approaches like the one followed in this study for rectal wall delineation [49]. In addition, we divided the rectal wall into three parts (upper 1/3, mid 1/3, and lower 1/3) because segments of the rectum/rectal wall situated beyond the radiation beam are subjected to low levels of radiation exposure.

The dosimetric analysis demonstrates noticeable variations between the rectum, rectal wall, and rectal wall subsegments in this study. For all dose endpoints, the rectal wall volumes were increased compared to rectum volumes, and there was a significant difference in all; however, all values were within the limits of QUANTEC dose-volume constraints.

For all dose endpoints, the upper 1/3 rectal volumes were statistically significantly lower than the rectum volumes. Regarding the mid 1/3 rectal volumes, we found a significant increase compared to the rectum. It is noteworthy that the dose-volume constraint of 56.52 Gy (V56.52_Gy_ = 29.4%) exceeds the QUANTEC constraint, while all other dosimetric values were within limits, presumably because the mid-segment of the rectal wall is always within the irradiated volume.

The lower 1/3 rectal volumes were increased compared to rectum volumes; for dose endpoints of 52.17 Gy and 56.52 Gy statistically difference was observed. All lower 1/3 rectal volumes were within the QUANTEC constraints. Furthermore, it was not possible to establish a correlation between dosimetry and late rectal toxicity due to the limited number of individuals who experienced adverse effects.

The correlation between dosimetric factors and the incidence of late rectal toxicity in the context of MHRT has been the subject of prior research. Akimoto et al. investigated the probable correlation between clinical and dosimetric parameters and the incidence of rectal bleeding grade 2 or higher in patients treated with hypofractionated 3DCRT for prostate cancer [18]. The study involved 52 patients who received a total dose of 69 Gy, administered in thrice-weekly fraction doses of 3 Gy. The median time until Grade 2 or more severe rectal bleeding was found to be 11 months, similar to our finding. The univariate analysis indicated that diabetes mellitus (*p* < 0.001) and rectal dose-volume parameters of V30_Gy_ > 60%, V50_Gy_ > 40% (*p* < 0.05), V80_Gy_ > 25%, and V90_Gy_ > 15% (*p* < 0.001), were statistically significant factors, associated with an increased risk of rectal bleeding grade 2 or higher. However, after multivariate analysis, diabetes was the most significant risk factor for rectal bleeding (*p* < 0.05) rather than rectal dose-volume parameters.

Faria et al. conducted a study on 71 patients who had favorable risk prostate adenocarcinoma and underwent an MHRT treatment schedule (66 Gy, 3 Gy per fraction) with 3DCRT.Their analysis derived from the actual DVH dose parameters for the rectum revealed no correlation between the rectum DVH parameters and late rectal toxicity [19].

The study by Vespiri et al. included a larger number of patients (*n* = 121) and utilized the same hypofractionation scheme as Faria et al., though engaging the IMRT technique [20]. They found that rectal wall DVH parameters of Dmax (*p* < 0.001), V36_Gy_ (*p* < 0.001), V48_Gy_ (*p* < 0.001), and V60_Gy_ (*p* = 0.001) were associated with late GI toxicity.

Thor et al. investigated dose-volume predictors of late GI adverse events in the MHRT subgroup of patients from the RTOG 0415 phase III [22]. For MHRT schedules like in the RTOG 0415 study (70 Gy, 2.5 Gy per fraction), it is recommended that 5% of the prescribed dose (D5%) be maintained at or below 62 Gy to reduce late GI toxicity from 20% to 10%. The rectum constraints outlined in the radiotherapy protocol of the RTOG 0415 study were as follows: V74_Gy_ < 15%, V69_Gy_ < 25%, V64_Gy_ < 35% and V59_Gy_ < 50% (data presented are EQD2 using α/β ratio 3 Gy) [9]. Sanguineti et al. conducted a retrospective study with the aim of verifying and potentially improving the dose volume objective for the rectal wall following MHRT delivered either with EDCRT or IMRT (62 Gy, 3.1 Gy per fraction four times per week) [16]. The planning objectives for doses to the rectal wall were V38 ≤ 50%, V54 ≤ 30%, and Dmax (0.035 cc) for 3DCRT. Using data from DVH, in 106 patients treated with 3DCRT, they were unable to identify independent dosimetric predictors of late rectal hemorrhage in their analysis. Dolobel et al. attempted to provide a nomogram for individual prediction of late rectal toxicity considering the three following parameters: radiotherapy technique, dose escalation, and the utilization of MHRT. They did not include planning parameters such as dose-volume histograms [21].

Despite the fact that all of the aforementioned studies were conducted within the setting of MHRT, various MHRT schedules were utilized (69 Gy administered in three-Gy fraction doses per week as opposed to 66 Gy with 3 Gy per fraction or 70 Gy, 2.5 Gy per fraction), and distinct radiotherapy techniques (IMRT as opposed to 3DCRT) or structures (rectum as opposed to rectal wall) were applied to the analysis of DVH parameters.

Furthermore, even though data presented by Thor et al. was derived from a RCT phase III study (RTOG 0415), their results are based on non-QUANTEC dosimetric constraints, which makes a direct comparison non-possible.

This study found that rectal mucosal injury related to radiotherapy occurred 12 months post-treatment, and even if the grade of rectal mucosal damage according to VRS was changed, no new patients were added in the subsequent 12-month period. This is also noted in the study by Van Lin et al., where at 3, 6, 12, and 24 months after RT proctoscopy on 48 patients with prostate cancer found that grade 2 and 3 telangiectasia were stable, whereas grade 1 spontaneously resolved [50]. Hence, the rectal mucosal damage can be regarded as a delayed event of radiotherapy.

Despite rectal mucosal injury being a delayed event, a correlation between our VRS findings and late GI toxicity could not be established. This could be attributed to the restricted number of patients.

Our results support the suggestion of Ippolito et al. performing a rectoscopy at 12 months after radiotherapy in patients treated with radical MHRT, delivered with 3DCRT, since the rectoscopy performed at 12 months after radiotherapy detected rectal mucosal damage in 35% of the patients [27]. Ippolito et al. furthermore suggested that a rectoscopy one year after conventional fractionated radiotherapy could predict delayed rectal toxicity. However, the study’s limitations include population variability in treatment intents, volumes, doses, technique, and uncertainty in the initial state of the rectum due to the absence of an endoscopic examination before radiotherapy.

The findings of our study compared to rectoscopy findings at 12 months post-treatment showed that patients with rectal mucosal injury had significantly increased irradiated volume of the rectum at dose endpoints of V34.78_Gy_, V43.48_Gy_ V52.17_Gy_, and V56.52_Gy_ compared to patients with normal rectal mucosa. Despite the increased irradiated volume, the values were within the QUANTEC limits.

Rectal wall showed significantly increased irradiated volume in patients with rectal mucosal damage in the same endpoints as in rectum.

The mid 1/3 of the rectal wall revealed a significant increase for the dose endpoint of V60.87_Gy_ for patients with rectal damage compared to the ones without, but the increase was kept within the QUANTEC limits. Interestingly V52.17_Gy_, and V56.52_Gy_ of the mid 1/3 of the rectal wall were exceeding the QUANTEC limits for all patients since the mid segment of the rectal wall is within the irradiated volume.

The lower 1/3 rectal wall volume showed a significant increase for dose endpoints of V34.78_Gy_, V43.48_Gy_, V52.17_Gy_, V56.52_Gy_, and V60.87_Gy_. The increase was over the QUANTEC limits for the endpoints of V43.48_Gy_, V52.17_Gy_, and V56.52_Gy,_ as the lower segment of the rectal wall is within the irradiated volume. This study underlines the fact that the rectal wall is a more sensitive structure compared to the rectum in terms of the potential of rectal mucosal damage.

In a subsequent analysis by Ippolito et al., early rectal mucosal damage was associated with rectum dosimetric parameters [26]. The strongest dosimetric indicators for predicting grade 2 telangiectasia were rectal V60_Gy_ (*p* = 0.014), rectal V70_Gy_ (*p* = 0.017), and rectal Dmean (*p* = 0.018). Additionally, comparable findings were observed for grade 2 VRS. The combination of V60_Gy_ < 34.4%, V70_Gy_ < 16.7%, and Dmean < 57.5 Gy was linked to a reduced likelihood of grade 2 telangiectasia and VRS. Our results align with those of Ippolito et al. concerning endpoints at higher doses and rectal mucosal injury.

Available data is limited in the setting of MHRT regarding DVH parameters related to rectoscopy findings. A study performed (65 Gy/2.6 Gy per fraction) with VMAT technique, followed by a radiosurgery boost showed that radiosurgery boost is an independent risk factor for developing rectal mucosal damage. However, no dosimetric parameters were included in this study [28].

This work presents results regarding dosimetric parameters and endoscopic outcomes in the setting of MHRT for intermediate-risk cancer patients adopting 3DCRT. This analysis presents evidence supporting the hypothesis that the rectal wall is a more sensitive critical structure as opposed to the rectum when 3DCRT is used and satisfying the QUANTEC constrains in the rectal wall may lead to decrease of late GI toxicity.

Furthermore, it was observed that the mid and lower segments of the rectal wall are within the irradiated volume when the prostate and seminal vesicles are the targets of treatment. Therefore, a more precise delineation of these anatomical components could potentially reduce late rectal toxicity. In addition, our findings contribute to the existing body of literature by implying that rectoscopy performed 12 months after radiotherapy could function as a surrogate endpoint for the early detection of late rectal toxicity. V52.17_Gy_ and V56.52_Gy_ were the common dosimetric parameters that were significant for rectal mucosal injury for the rectum, rectal wall, mid and lower 1/3 rectal wall. It is our proposition that these dosimetric parameters could potentially function as predictors of rectal mucosal injury in the setting of MHRT. However, we could not establish a correlation between late GI toxicity and endoscopic findings or dosimetric parameters, possibly due to the small number of patients who presented late adverse events.

The prospective design, the homogeneity of the included population with respect to the hypofractionated radiotherapy schedule (all patients undergo the same MHRT scheme), the disease-specific characteristics (all patients have intermediate risk features), and the incorporation of longitudinal endoscopy data from a routine clinical setting can be considered as strengths of this study.

The limited sample size of our patient population and the absence of toxicity data beyond the two-year mark, which precludes the evaluation of delayed complications, are considered limitations of the study. GI late toxicity and rectal mucosal injury following radiotherapy may also be associated with patient-specific variables that were not considered in our analysis, such as age, diabetes, and anticoagulant medication.

## 5. Conclusions

In prostate cancer patients with intermediate-risk characteristics, MHRT delivered with 3DCRT and QUANTEC constraints modified according to the LQ model, with an α/β ratio of 3 Gy for late GI toxicity, can be considered both feasible and safe. In addition to the rectum, a thorough delineation of the rectal wall and its subsegments, as well as a dosimetric analysis of these structures, may reduce late GI adverse events. Dosimetric parameters such as V52.17_Gy_ and V56.52_Gy_ were identified to have a significant impact on rectal mucosal injury; however, additional dose endpoint(s) validation and its relation to late GI toxicity in a large patient cohort could provide further insight into the relationship between hypofractionated radiotherapy and rectal toxicity, especially in today’s clinical practice where MHRT seems to be the new standard of care. Furthermore, the implementation of these findings to more advanced radiotherapy techniques such as IMRT or VMAT, which demonstrate a clear advantage in dose coverage, conformity, and homogeneity over 3DCRT, may lead to reduced rectal toxicity. Similar works in the future could lead toward establishing a correlation between dose endpoints and the absence of late GI adverse events, especially in the IMRT and VMAT era, where doses to the normal tissues have been reduced but not eliminated.

## Figures and Tables

**Figure 1 cancers-16-01192-f001:**
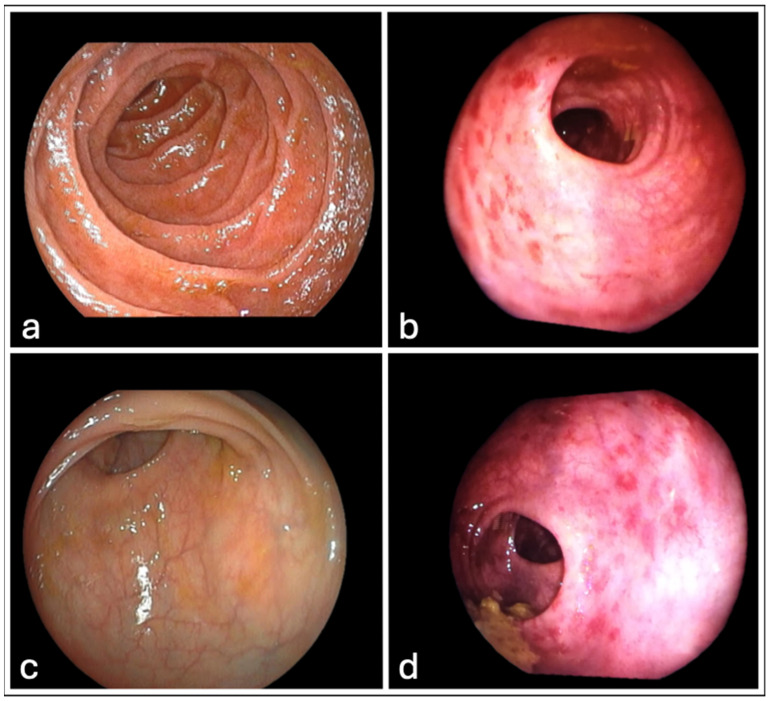
Typical images from two patients undergone rectoscopy. (**a**,**c**) display normal rectal mucosal before starting radiotherapy. (**b**,**d**) show the same patients 12 months after receiving radiotherapy, displaying rectal mucosal injury VRS1.

**Figure 2 cancers-16-01192-f002:**
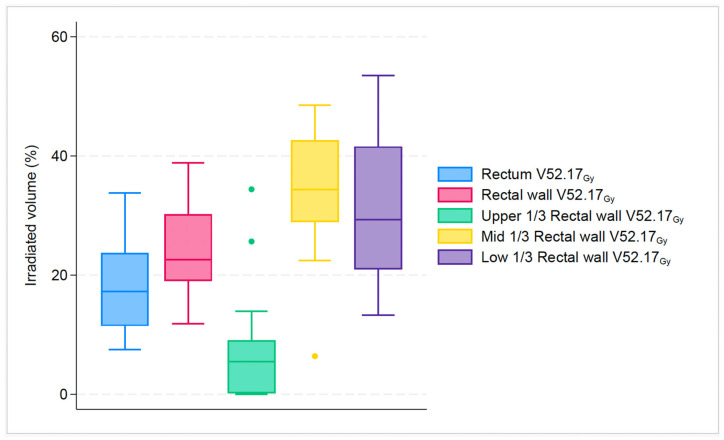
Boxplot graph that shows the median percentage along with 25th, 75th percentiles, of irradiated volume between rectum, rectal wall and its subsegments for the specific dose endpoint of V52.17_Gy_. The dots represent the outliers.

**Figure 3 cancers-16-01192-f003:**
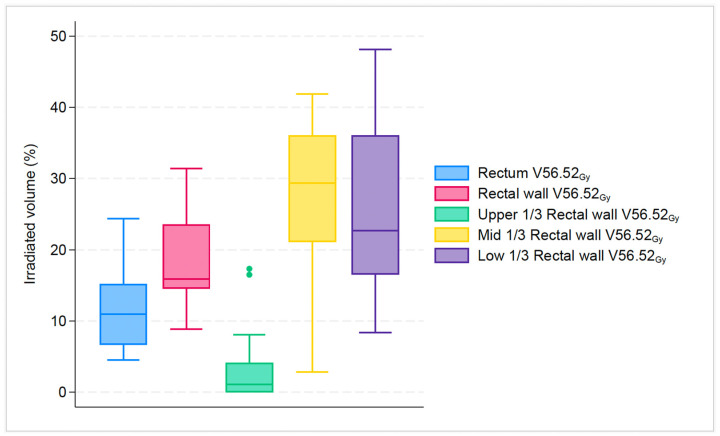
Boxplot graph that shows the median percentage along with 25th, 75th percentiles, of irradiated volume between rectum, rectal wall and its subsegments for the specific dose endpoint of V56.52_Gy_. The dots represent the outliers.

**Table 1 cancers-16-01192-t001:** Patients’ characteristics.

Characteristic	N (%)
Age at diagnosis *	72.7 years (6.3)
Histology	
adenocarcinoma	20 (100)
Clinical TNM	
T1cN0M0	5 (25)
T2aN0M0	3 (15)
T2bN0M0	4 (20)
T2cN0M0	8 (40)
PSA at diagnosis *	7.4 (12.6)
Gleason Score	
6 (3 + 3)	5 (25)
7 (3 + 4)	10 (50)
7 (4 + 3)	5 (25)
Grade	
1	5 (25)
2	10 (50)
3	5 (25)
4	0 (0)
5	0 (0)
Adjuvant ADT	20 (100)
Duration of ADT (months)	6 (0)
Diabetes	
No	13 (65)
Yes	7 (35)
Anti-coagulant medication	
No	13 (65)
Yes	7 (35)
Non malignant findings of the colonoscopy	
No	10 (50)
Yes	10 (50)
Pre-treatment symptoms from GI	
No	20 (100)
Yes	0 (0)
Pre-treatment symptoms from GU,	
No	No 14(70)
Yes	Yes 6 (30)
Completed protocol schedule (yes or no)	All received
Interruptions during RT	
No	20 (100)
Yes	0 (0)
CTV Prostate Volume * (cm^3^)	47.6 (23.4)

* median (interquartile range). Abbreviations: PSA, prostate-specific antigen; ADT, androgen deprivation therapy; GI, gastrointestinal; GU, genitourinary.

**Table 2 cancers-16-01192-t002:** Acute GI-adverse events.

GI Early Adverse Events	Grade 0 N (%)	Grade 1 N (%)	Grade 2 N (%)	Grade 3 N (%)	Grade 4 N (%)
1st week of RT	20 (100)	0 (0)	0 (0)	0 (0)	0 (0)
2nd week of RT	20 (100)	0 (0)	0 (0)	0 (0)	0(0)
3rd week of RT	13 (65)	6 (30)	1 (5)	0 (0)	0 (0)
4th week of RT	15 (70)	3 (15)	2 (10)	0 (0)	0 (0)
End of RT	18 (90)	2 (10)	0 (0)	0 (0)	0 (0)
3 months post- RT	20 (100)	0 (0)	0 (0)	0 (0)	0 (0)
6 months post- RT	20 (100)	0 (0)	0 (0)	0 (0)	0 (0)

Abbreviations: GI, gastrointestinal; N, number; RT, radiotherapy.

**Table 3 cancers-16-01192-t003:** Acute GU-adverse events.

GU Early Adverse Events	Grade 0 N (%)	Grade 1 N (%)	Grade 2 N (%)	Grade 3 N (%)	Grade 4 N (%)
1st week of RT	19 (95)	1 (5)	0 (0)	0 (0)	0 (0)
2nd week of RT	18 (90)	2 (10)	0 (0)	0 (0)	0 (0)
3rd week of RT	12 (60)	8 (40)	0 (0)	0 (0)	0 (0)
4th week of RT	9 (45)	11 (55)	0 (0)	0 (0)	0 (0)
End of RT	10 (50)	10 (50)	0 (0)	0 (0)	0 (0)
3 months post-RT	20 (100)	0 (0)	0 (0)	0 (0)	0 (0)
6 months post-RT	19 (95)	1 (5)	0 (0)	0 (0)	0 (0)

Abbreviations: GU, genitourinary; N, number; RT, radiotherapy.

**Table 4 cancers-16-01192-t004:** Late adverse events GI.

GI Late Adverse Events	Grade 0 N (%)		Grade 1 N (%)		Grade 2 N (%)		Grade 3 N (%)		Grade 4 N (%)	
Scale	LENT/SOMA	S-RS	LENT/SOMA	S-RS	LENT/SOMA	S-RS	LENT/SOMA	S-RS	LENT/SOMA	S-RS
9 months post-RT	20 (100)	20 (100)	0 (0)	0 (0)	0 (0)	0 (0)	0 (0)	0 (0)	0 (0)	0 (0)
12 months post-RT	18 (90)	18 (90)	1 (5)	1 (5)	0 (0)	0 (0)	1 (5)	1 (5)	0 (0)	0 (0)
18 months post-RT	19 (95)	19 (95)	0 (0)	0 (0)	1 (5)	1 (5)	0 (0)	0 (0)	0 (0)	0 (0)
24 months post-RT	19 (95)	19 (95)	0 (0)	0 (0)	1 (5)	1 (5)	0 (0)	0 (0)	0 (0)	0 (0)

Abbreviations: GI, gastrointestinal; N, number; LENT/SOMA, Late Effects Normal Tissue Task Force/Subjective, Objective, Management, Analytic; S-RS, Subjective-RectoSigmoid; RT, radiotherapy.

**Table 5 cancers-16-01192-t005:** Late adverse events GU.

GU Late Adverse Events	Grade 0 N (%)	Grade 1 N (%)	Grade 2 N (%)	Grade 3 N (%)	Grade 4 N (%)
9 months post-RT	19 (95)	1 (5)	0 (0)	0 (0)	0 (0)
12 months post-RT	19 (95)	1 (5)	0 (0)	0 (0)	0 (0)
18 months post-RT	18 (90)	2 (10)	0 (0)	0 (0)	0 (0)
24 months post-RT	14 (70)	6 (30)	0 (0)	0 (0)	0 (0)

Abbreviations: GU, genitourinary; N, number; RT, radiotherapy.

**Table 6 cancers-16-01192-t006:** Rectoscopy findings.

Rectoscopy	VRS 0 N (%)	VRS 1 N (%)	VRS 2 N (%)	VRS 3 N (%)
3 months post-RT	20 (100)	0 (0)	0 (0)	0 (0)
6 months post-RT	18 (90)	0 (0)	2 (10)	0 (0)
9 months post-RT	18 (90)	0 (0)	2 (10)	0 (0)
12 months post-RT	13 (65)	4 (20)	2 (10)	1 (5)
18 months post-RT	13 (65)	4 (20)	3 (15)	0 (0)
24 months post-RT	13 (65)	4 (20)	3 (15)	0 (0)

Abbreviations: VRS, Vienna Rectoscopy Score; N, number; RT, radiotherapy.

**Table 7 cancers-16-01192-t007:** Late GI-related adverse events according to the LENT-SOMA scale and the rectoscopy findings evaluated according to the VRS.

Months after RT Colonoscopy		LENT-SOMA Scale	VRS	*p*-Value
	Discrepancies N (%)			
3	0 (0)			-
6	2 (10)	0 0	2 2	0.16
9	2 (10)	0 0	2 2	0.16
12	5 (25)	0 0 0 0 0	1 1 1 2 2	0.063
18	6 (30)	0 0 0 0 0 0	1 1 1 1 2 2	0.031
24	6 (30)	0 0 0 0 0 0	1 1 1 1 2 2	0.031

Abbreviations: RT, radiotherapy; LENT/SOMA, Late Effects Normal Tissue Task Force/Subjective, Objective, Management, Analytic; VRS, Vienna Rectoscopy Score; N, number.

**Table 8 cancers-16-01192-t008:** Dose-Volume constraints of rectum according to QUANTEC and the EQD2 for late toxicity (α/β = 3).

QUANTEC dose Constrains for Rectum	EQD2 for Late Toxicity (α/β = 3).
V30_Gy_	V26.08_Gy_
V40_Gy_	V34.78_Gy_
V50_Gy_ < 50%	V43.48_Gy_ < 50%
V60_Gy_ < 35%	V52.17_Gy_ < 35%
V65_Gy_ < 25%	V56.52_Gy_ < 25%
V70_Gy_ < 20%	V60.87_Gy_ < 20%

Note: there are no constrains for dose volume below 50 Gy according to QUANTEC. Abbreviation: QUANTEC, Quantitative Analyzes of Normal Tissue Effects in the Clinic; EQD2, Equivalent Dose in 2Gy Fractions; V, volume.

**Table 9 cancers-16-01192-t009:** Dosimetry data of the rectum correlated to rectoscopy findings according to VRS.

Rectoscopy Findings	V26.08_Gy_	V34.78_Gy_	V43.48_Gy_	V52.17_Gy_	V56.52_Gy_	V60.87_Gy_
3 months post-RT						
Group A	89.2 (10.5)	48.5 (28.8)	32.3 (22.4)	17.2 (12.3)	11 (8.6)	0.2 (2.3)
Group B	-	-	-	-	-	-
*p* value	-	-	-	-	-	-
6 months post-RT						
Group A	89.2 (15)	47 (25.5)	30.9 (20.4)	15.8 (12.3)	9.5 (7.7)	0.08 (2.1)
Group B	88.7 (10.5)	68.8 (1.6)	48.6 (1.2)	26.1 (15.3)	18 (12)	3 (5.1)
*p* value	0.85	0.042	0.042	0.26	0.26	0.32
9 months post-RT						
Group A	89.2 (15)	47 (25.5)	30.9 (20.4)	15.8 (12.3)	9.5 (7.7)	0.08 (2.1)
Group B	88.7 (10.5)	68.8 (1.6)	48.6 (1.2)	26.1 (15.3)	18 (12)	3 (5.1)
*p* value	0.85	0.042	0.042	0.26	0.26	0.32
12 months post-RT						
Group A	87 (25.6)	40 (17.7)	27.6 (13)	11.8 (8.7)	7.4 (7.3)	0.002 (0.2)
Group B	94 (12.2)	61.8 (10)	45.6 (10.5)	27.8 (14.2)	20.7 (12.4)	2.1 (5.1)
*p* value	0.19	0.009	0.006	0.006	0.006	0.052
18 months post-RT						
Group A	87 (25.6)	40 (17.7)	27.6 (13)	11.8 (8.7)	7.4 (7.3)	0.002 (0.2)
Group B	94 (12.2)	61.8 (10)	45.6 (10.5)	27.8 (14.2)	20.7 (12.4)	2.1 (5.1)
*p* value	0.19	0.009	0.006	0.006	0.006	0.052
24 months post-RT						
Group A	87 (25.6)	40 (17.7)	27.6 (13)	11.8 (8.7)	7.4 (7.3)	0.002 (0.2)
Group B	94 (12.2)	61.8 (10)	45.6 (10.5)	27.8 (14.2)	20.7 (12.4)	2.1 (5.1)
*p* value	0.19	0.009	0.006	0.006	0.006	0.052

Note: the values presented are in median (interquartile range); Group A is referred to patients within normal limits of VRS; Group B is referred to patients with any VRS rectal mucosal damage. Abbreviations: VRS, Vienna Rectoscopy Score; V, volume; RT, radiotherapy.

**Table 10 cancers-16-01192-t010:** Dosimetry data of the rectal wall correlated to rectoscopy findings, according to VRS.

Coloscopy Findings	V26.08_Gy_	V34.78_Gy_	V43.48_Gy_	V52.17_Gy_	V56.52_Gy_	V60.87_Gy_
3 months after RT						
Group A	90 (13.6)	49.6 (16)	39.2 (12.4)	22.6 (11.2)	15.9 (9)	1.3 (8)
Group B	-	-	-	-	-	-
*p* value	-	-	-	-	-	-
6 months after RT						
Group A	90 (14.2)	46.4 (15.2)	36.7 (12.6)	21.6 (11.1)	15.9 (8.4)	0.6 (7.8)
Group B	88.1 (9.5)	59.1 (2.9)	44.7 (4.9)	30.2 (14.9)	23.3 (16.2)	8.1 (12.8)
*p* value	0.85	0.13	0.17	0.32	0.52	0.26
9 months after RT						
Group A	90 (14.2)	46.4 (15.2)	36.7 (12.6)	21.6 (11.1)	15.9 (8.4)	0.6 (7.8)
Group B	88.1 (9.5)	59.1 (2.9)	44.7 (4.9)	30.2 (14.9)	23.3 (16.2)	8.1 (12.8)
*p* value	0.85	0.13	0.17	0.32	0.52	0.26
12 months after RT						
Group A	89.5 (14.6)	42.8 (12.4)	33.2 (8.8)	19.9 (4.2)	15.7 (3.3)	0.003 (1.4)
Group B	92.8 (12.7)	55.5 (6.5)	42.3 (5.8)	30.9 (14.7)	24.7 (15.8)	7.8 (8.9)
*p* value	0.29	0.005	0.005	0.005	0.046	0.033
18 months after RT						
Group A	89.5 (14.6)	42.8 (12.4)	33.2 (8.8)	19.9 (4.2)	15.7 (3.3)	0.003 (1.4)
Group B	92.8 (12.7)	55.5 (6.5)	42.3 (5.8)	30.9 (14.7)	24.7 (15.8)	7.8 (8.9)
*p* value	0.29	0.005	0.005	0.005	0.046	0.033
24 months after RT						
Group A	89.5 (14.6)	42.8 (12.4)	33.2 (8.8)	19.9 (4.2)	15.7 (3.3)	0.003 (1.4)
Group B	92.8 (12.7)	55.5 (6.5)	42.3 (5.8)	30.9 (14.7)	24.7 (15.8)	7.8 (8.9)
*p* value	0.29	0.005	0.005	0.005	0.046	0.033

Note: the values presented are in median (interquartile range); Group A is referred to patients within normal limits of VRS; Group B is referred to patients with any VRS rectal mucosal damage. Abbreviations: VRS, Vienna Rectoscopy Score; V, volume; RT, radiotherapy.

**Table 11 cancers-16-01192-t011:** Dosimetry data of the upper 1/3 of the rectal wall correlated to rectoscopy findings, according to VRS.

Coloscopy Findings	V26.08_Gy_	V34.78_Gy_	V43.48_Gy_	V52.17_Gy_	V56.52_Gy_	V60.87_Gy_
3 months post-RT						
Group A	78.4 (39.1)	36.4 (23.6)	21.7 (23.2)	5.5 (8.9)	1.1 (4.1)	0 (0)
Group B	-	-	-	-	-	-
*p* value	-	-	-	-	-	-
6 months post-RT						
Group A	83.5 (32.3)	36.4 (24.1)	24.1 (25.2)	5.5 (9.6)	0.8 (5.6)	0 (0)
Group B	60.1 (33.7)	32.8 (19.9)	17.7 (7)	4.3 (4.7)	1.7 (2)	0 (0)
*p* value	0.38	0.59	0.44	>0.99	0.61	>0.99
9 months post-RT						
Group A	83.5 (32.3)	36.4 (24.1)	24.1 (25.2)	5.5 (9.6)	0.8 (5.6)	0 (0)
Group B	60.1 (33.7)	32.8 (19.9)	17.7 (7)	4.3 (4.7)	1.7 (2)	0 (0)
*p* value	0.38	0.59	0.44	>0.99	0.61	>0.99
12 months post-RT						
Group A	79.9 (21.9)	35.8 (19.4)	22.3 (12.9)	5.4 (8.4)	0 (5.6)	0 (0)
Group B	76.9 (53.2)	42.8 (23.6)	21.2 (33.7)	5.5 (8.9)	1.5 (2.6)	0 (0)
*p* value	0.8	0.7	0.7	0.53	0.8	>0.99
18 months post-RT						
Group A	79.9 (21.9)	35.8 (19.4)	22.3 (12.9)	5.4 (8.4)	0 (5.6)	0 (0)
Group B	76.9 (53.2)	42.8 (23.6)	21.2 (33.7)	5.5 (8.9)	1.5 (2.6)	0 (0)
*p* value	0.8	0.7	0.7	0.53	0.8	>0.99
24 months post-RT						
Group A	79.9 (21.9)	35.8 (19.4)	22.3 (12.9)	5.4 (8.4)	0 (5.6)	0 (0)
Group B	76.9 (53.2)	42.8 (23.6)	21.2 (33.7)	5.5 (8.9)	1.5 (2.6)	0 (0)
*p* value	0.8	0.7	0.7	0.53	0.8	>0.99

Note: the values presented are in the median (interquartile range); Group A refers to patients within normal limits of VRS; Group B refers to patients with any VRS rectal mucosal damage. Abbreviations: VRS, Vienna Rectoscopy Score; V, volume; RT, radiotherapy.

**Table 12 cancers-16-01192-t012:** Dosimetry data of the mid 1/3 of the rectal wall correlated to rectoscopy findings, according to VRS.

Coloscopy Findings	V26.08_Gy_	V34.78_Gy_	V43.48_Gy_	V52.17_Gy_	V56.52_Gy_	V60.87_Gy_
3 months post-RT						
Group A	100 (0)	58 (13.3)	46.3 (10.6)	34.4 (13.7)	29.4 (15)	0.4 (3.3)
Group B	-	-	-	-	-	-
*p* value	-	-	-	-	-	-
6 months post-RT						
Group A	100 (0)	54.6 (13.7)	45 (10.9)	34.4 (11.8)	29.4 (14.2)	0.07 (3.2)
Group B	100 (0)	66.4 (9.3)	52.4 (11.4)	35.5 (26.1)	24.9 (34)	10.2 (14)
*p* value	>0.99	0.095	0.26	0.95	0.95	0.13
9 months post-RT						
Group A	100 (0)	54.6 (13.7)	45 (10.9)	34.4 (11.8)	29.4 (14.2)	0.07 (3.2)
Group B	100 (0)	66.4 (9.3)	52.4 (11.4)	35.5 (26.1)	24.9 (34)	10.2 (14)
*p* value	>0.99	0.095	0.26	0.95	0.95	0.13
12 months post-RT						
Group A	100 (0)	52.3 (14.7)	41.8 (9.6)	32.5 (9.6)	28 (9.4)	0 (2)
Group B	100 (0.1)	61.2 (14.3)	48.6 (11.6)	39.9 (14.3)	32.4 (17.5)	3.1 (14.7)
*p* value	0.22	0.056	0.16	0.24	0.24	0.034
18 months post-RT						
Group A	100 (0)	52.3 (14.7)	41.8 (9.6)	32.5 (9.6)	28 (9.4)	0 (2)
Group B	100 (0.1)	61.2 (14.3)	48.6 (11.6)	39.9 (14.3)	32.4 (17.5)	3.1 (14.7)
*p* value	0.22	0.056	0.16	0.24	0.24	0.034
24 months post-RT						
Group A	100 (0)	52.3 (14.7)	41.8 (9.6)	32.5 (9.6)	28 (9.4)	0 (2)
Group B	100 (0.1)	61.2 (14.3)	48.6 (11.6)	39.9 (14.3)	32.4 (17.5)	3.1 (14.7)
*p* value	0.22	0.056	0.16	0.24	0.24	0.034

Note: the values presented are in median (interquartile range); Group A is referred to patients within normal limits of VRS; Group B is referred to patients with any VRS rectal mucosal damage. Abbreviations: VRS, Vienna Rectoscopy Score; V, volume; RT, radiotherapy.

**Table 13 cancers-16-01192-t013:** Dosimetry data of the lower 1/3 of the rectal wall correlated to rectoscopy findings, according to VRS.

Coloscopy Findings	V26.08_Gy_	V34.78_Gy_	V43.48_Gy_	V52.17_Gy_	V56.52_Gy_	V60.87_Gy_
3 months post-RT						
Group A	99.5 (14.8)	49.8 (19.3)	38.6 (17.2)	29.3 (20.6)	22.7 (19.6)	0.4 (8.2)
Group B	-	-	-	-	-	-
*p* value	-	-	-	-	-	-
6 months post-RT						
Group A	98.7 (14.8)	48 (20.8)	37.4 (17.5)	28.1 (15.5)	21.2 (14.3)	0.3 (7.2)
Group B	100 (0)	69.8 (23.6)	55.7 (15.5)	47.1 (10.8)	40.9 (9.1)	12.1 (23.9)
*p* value	0.3	0.095	0.063	0.095	0.095	0.37
9 months post-RT						
Group A	98.7 (14.8)	48 (20.8)	37.4 (17.5)	28.1 (15.5)	21.2 (14.3)	0.3 (7.2)
Group B	100 (0)	69.8 (23.6)	55.7 (15.5)	47.1 (10.8)	40.9 (9.1)	12.1 (23.9)
*p* value	0.3	0.095	0.063	0.095	0.095	0.37
12 months post-RT						
Group A	98.3 (14.8)	44.7 (17.4)	33.6 (12.9)	26.8 (9)	18.6 (12.2)	0 (1.7)
Group B	100 (3.1)	62.5 (21.7)	52.3 (21.5)	44.5 (22.7)	36.4 (22.4)	9.1 (23.9)
*p* value	0.19	0.005	0.002	0.003	0.006	0.015
18 months post-RT						
Group A	98.3 (14.8)	44.7 (17.4)	33.6 (12.9)	26.8 (9)	18.6 (12.2)	0 (1.7)
Group B	100 (3.1)	62.5 (21.7)	52.3 (21.5)	44.5 (22.7)	36.4 (22.4)	9.1 (23.9)
*p* value	0.19	0.005	0.002	0.003	0.006	0.015
24 months post-RT						
Group A	98.3 (14.8)	44.7 (17.4)	33.6 (12.9)	26.8 (9)	18.6 (12.2)	0 (1.7)
Group B	100 (3.1)	62.5 (21.7)	52.3 (21.5)	44.5 (22.7)	36.4 (22.4)	9.1 (23.9)
*p* value	0.19	0.005	0.002	0.003	0.006	0.015

Note: the values presented are in median (interquartile range); Group A is referred to patients within normal limits of VRS; Group B is referred to patients with any VRS rectal mucosal damage. Abbreviations: VRS, Vienna Rectoscopy Score; V, volume; RT, radiotherapy.

## Data Availability

Data is unavailable, but it will be provided upon request by the correspondence author.

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
