# Peer review of "Comprehensive 3DCRT Hypofractionated Radiotherapy Schedule for Localized Prostate Adenocarcinoma in the Era of IMRT: Dosimetric and Endoscopic Analysis"

_cancers, 2024, doi:10.3390/cancers16061192_

Round 1
Reviewer 1 Report
Comments and Suggestions for Authors
Dear Authors
ID
Cancers-2874968
Comprehensive 3DCRT Hypo-fractionated Radiotherapy Schedule for Localized Prostate Adenocarcinoma in the Era of IMRT: Dosimetric and Endoscopic Analysis by Kougioumtzopoulou et al summarizes In, prostate cancer patients with intermediate risk characteristics MHRT delivered with 3DCRT and QUANTEC constraints modified according to the LQ model, with a 540 α/β ratio of 3 Gy for late GI toxicity, can be considered both feasible and safe. In addition to the rectum, a thorough delineation of the rectal wall and its subsegments, as well as dosimetric analysis of these structures may reduce late GI adverse events. Dosimetric parameters such as V52.17Gy and V56.52Gy were identified to have a significant impact to rectal mucosal injury; however, additional dose endpoint(s) validation and its relation to late GI toxicity in a large patient cohort could provide a further insight to the relationship between hypo-fractionated radiotherapy and rectal toxicity, especially in today’s clinical practice where MHRT seems to be the new standard of care. Furthermore, the implementation of these findings to more advanced radiotherapy techniques, such as IMRT or VMAT which demonstrate a clear advantage in dose coverage, conformity, and homogeneity over 3DCRT may lead to reduced rectal toxicity.
This is an interesting article; however, it needs some additional data to strengthens the manuscript
Comments:
1. What is the readout for GI toxicity?
2. Please provide some evidence (image) before after the radiation
Comments on the Quality of English Language
Minor edits requred
Author Response
On behalf of all co-authors, I would like to thank the reviewer for the comments.
Comments:
- What is the readout for GI toxicity?
Answer: If we understand correctly, the readout of GI toxicity was based on the use of a video colonoscope and the assessment of radiation induced epithelial damage followed the Vienna Rectoscopy Score (VRS).
In the revised manuscript we have added the following: “The endoscopy was performed using a video colonoscope.” In the materials and methods section 2.4 2.4. Toxicity and Follow up, lines 173-174.
- Please provide some evidence (image) before after the radiation
Answer: We added typical images of a study participant before radiotherapy and at 12 months post treatment. In results, 3.2.2. Late toxicity section “Typical rectoscopy images from two study patients before treatment and 12 months af-ter RT with a VRS 1 rectal mucosal lesion are shown in Figure 1.” lines 262-263. The figure is in the lines 296-299.

Reviewer 2 Report
Comments and Suggestions for Authors
Interesting prospective trial; the paper is well written and results are "in linea" with data from literature; I recommend acceptance of your work
Author Response
We thank the reviewer for the suggestion for acceptance of our work.
Reviewer 3 Report
Comments and Suggestions for Authors
The author highlighted hypo-fractionated radiotherapy for localized prostate cancer. The relationship between dosimetric and endoscopic findings was of great interest. The reviewer agrees with some of the content; however, the reviewer has proposed several suggestions to enhance the paper:
Major point
1. The authors should indicate the rationale for the prescription dose (44 Gy in 16 fractions to the seminal vesicles and to the prostate followed by a sequential boost to the prostate alone of 16.5 Gy in 6 fractions).
2. Treatment protocols should be clearly stated, including the use of neoadjuvant and/or adjuvant ADT.
3. The authors should provide details regarding grade 3 gastrointestinal (GI) toxicity at 12 months post-RT.
4. T The authors should illustrate the relationship between GI late adverse events (AE) and Vienna Rectoscopy Score (VRS) in a table or figure.
5. Finally, what level of V in this study's protocol would reduce GI late AE?
Minor point
1. Is the information presented at 3 months post-RT in Table 3 accurate?
Author Response
On behalf of all co-authors, I would like to thank the reviewer for the comments.
Major point
- The authors should indicate the rationale for the prescription dose (44 Gy in 16 fractions to the seminal vesicles and to the prostate followed by a sequential boost to the prostate alone of 16.5 Gy in 6 fractions).
Answer: We do agree with the reviewer. In the revised manuscript we have added the following: “The rationale to include the seminal vesicles in CTV1 was based on the Partin staging nomogram where the probability of seminal vesicle invasion is 15% for patients with one risk factor (PSA over 10, Gleason over 6, T-stage over T2a) and the prescription to the seminal vesicles and to the prostate followed the CHHip protocol for intermediate risk cancer [7,37]” References were added. In the materials and methods section 2.3 Radiotherapy, lines 133-137.
Reference:
- Dearnaley, D.; Syndikus, I.; Mossop, H.; Khoo, V.; Birtle, A.; Bloomfield, D.; Graham, J.; Kirkbride, P.; Logue, J.; Malik, Z.; Money-Kyrle, J.; O'Sullivan, J. M.; Panades, M.; Parker, C.; Patterson, H.; Scrase, C.; Staffurth, J.; Stockdale, A.; Tremlett, J.; Bid-mead, M.; Mayles, H.; Naismith, O.; South, C.; Gao, A.; Cruickshank, C.; Hassan, S.; Pugh, J.; Griffin, C.; Hall, E.; CHHiP Investiga-tors Conventional versus hypofractionated high-dose intensity-modulated radiotherapy for prostate cancer: 5-year outcomes of the randomised, non-inferiority, phase 3 CHHiP trial. Lancet Oncol. 2016, 17, 1047–1060.
- Eifler, J. B.; Feng, Z.; Lin, B. M.; Partin, M. T.; Humphreys, E. B.; Han, M.; Epstein, J. I.; Walsh, P. C.; Trock, B. J.; Partin, A. W. An updated prostate cancer staging nomogram (Partin tables) based on cases from 2006 to 2011. BJU International 2013, 111, 22–29.
- Treatment protocols should be clearly stated, including the use of neoadjuvant and/or adjuvant ADT.
Answer: We do agree with the reviewer. In the revised manuscript we have added the following: “The administration of neoadjuvant and/ or adjuvant androgen deprivation therapy (ADT) was implemented for all patients, for a short duration of 4-6 months, both preceding and throughout the course of radiotherapy.” In the materials and methods section 2.2 Patients, lines 116-118.
- The authors should provide details regarding grade 3 gastrointestinal (GI) toxicity at 12 months post-RT.
Answer: We do agree with the reviewer. The one patient with grade 3 GI toxicity after 12 months from the completion of the radiotherapy presented with rectal bleeding and hospitalization and minor interventions required. The following data were added to the section 3.2.2 Late toxicity “One patient experienced grade 3 gastrointestinal damage 12 months after completing radiotherapy, presenting with rectal bleeding that required hospitalisation and modest intervention.” In results section 3.2.2. Late Toxicity, lines 242-244.
- The authors should illustrate the relationship between GI late adverse events (AE) and Vienna Rectoscopy Score (VRS) in a table or figure.
Answer: We do agree with the reviewer. We provide the following table 7, lines 266 and 292-294
- Finally, what level of V in this study's protocol would reduce GI late AE?
Answer: Please note that this work includes several correlations between a level of volume and late GI AE. Actually, we have presented a number of correlations between volume and rectal mucosal damage which is an important aspect of the current work. Please see the following:
Depending on which structure is someone considering no toxicity was observed in (tables 9,10, 12 and 13):
- Considering the rectum 11.8% at dose endpoint V52.17Gy and 7.4% at dose endpoint V56.52Gy.
- Considering the rectal wall 19.9% at dose endpoint V52.17Gy and 15.7% at dose endpoint V56.52Gy.
- Considering the mid 1/3 of the rectal wall 32.5% at dose endpoint V52.17Gy and 28% at dose endpoint V56.52Gy.
- Considering the low 1/3 of the rectal wall 26.8 % at dose endpoint V52.17Gy and 18.6% at dose endpoint V56.52Gy.
Minor point
- Is the information presented at 3 months post-RT in Table 3 accurate?
Answer: Indeed, the reviewer comment is correct. We have corrected and update the data in Table 3. (see revised text)
Table 7: Late GI-related adverse events according to LENT-SOMA scale with the rectoscopy findings evaluated according to the VRS.
|
Months after RT Colonoscopy |
|
LENT-SOMA scale |
VRS |
p-value |
|
|
Discrepancies N (%) |
|
|
|
|
3 |
0 (0) |
|
|
- |
|
6 |
2 (10) |
0 0 |
2 2 |
.16 |
|
9 |
2 (10) |
0 0 |
2 2 |
.16 |
|
12 |
5 (25) |
0 0 0 0 0 |
1 1 1 2 2 |
.063 |
|
18 |
6 (30) |
0 0 0 0 0 0 |
1 1 1 1 2 2 |
.031 |
|
24 |
6 (30) |
0 0 0 0 0 0 |
1 1 1 1 2 2 |
.031 |

Reviewer 4 Report
Comments and Suggestions for Authors
The study has been well planned and nicely conducted.
Some suggestions to improve this manuscript are:
1. The figure legends require significant elaboration.
2. Please include future directions in the conclusion section.
Author Response
On behalf of all co-authors, I would like to thank the reviewer for the comments.
4th Reviewer
The figure legends require significant elaboration.
Answer: We do agree with the reviewer. Thus, all figure legends have been rephrased to
Figure 2. Boxplot graph that shows the median percentage along with 25th, 75th percentiles, of irradiated volume between rectum, rectal wall and its subsegments for the specific dose endpoint of V52.17Gy. The dots represent the outliers. Lines 329-331.
Figure 3. Boxplot graph that shows the median percentage along with 25th, 75th percentiles, of irradiated volume between rectum, rectal wall and its subsegments for the specific dose endpoint of V56.52Gy. The dots represent the outliers. Lines 333-335.
Please include future directions in the conclusion section.
Answer: we have updated the conclusion section to include the following
“Similar works in the future could lead towards establishing a correlation between dose endpoints and the absence of late GI adverse events, especially in the IMRT and VMAT era where doses to the normal tissues have been reduced but not eliminated.” In the Conclusion section, lines 575-578.

Round 2
Reviewer 3 Report
Comments and Suggestions for Authors
The aim of the study was to evaluate comprehensive 3DCRT hypofractionated radiotherapy schedule for localized prostate adenocarcinoma. Topic was interesting and the revision was well performed.
Reviewer 4 Report
Comments and Suggestions for Authors
The revised manuscript looks better.